# A New Network Structure for Speech Emotion Recognition Research

**DOI:** 10.3390/s24051429

**Published:** 2024-02-22

**Authors:** Chunsheng Xu, Yunqing Liu, Wenjun Song, Zonglin Liang, Xing Chen

**Affiliations:** School of Electronic Information Engineering, Changchun University of Science and Technology, Changchun 130022, China; xuchunsheng@mails.cust.edu.cn (C.X.); sy_883849@163.com (W.S.); zonglin.liang@foxmail.com (Z.L.); zhang1545479445@126.com (X.C.)

**Keywords:** speech emotion recognition, spectrograms, multi-head attention, Bi-GRU

## Abstract

Deep learning promotes the breakthrough of emotion recognition in many fields, especially speech emotion recognition (SER). As an important part of speech emotion recognition, the most relevant acoustic feature extraction has always attracted the attention of existing researchers. Aiming at the problem that the emotional information contained in the current speech signals is distributed dispersedly and cannot comprehensively integrate local and global information, this paper presents a network model based on a gated recurrent unit (GRU) and multi-head attention. We evaluate our proposed emotion model on the IEMOCAP and Emo-DB corpora. The experimental results show that the network model based on Bi-GRU and multi-head attention is significantly better than the traditional network model at detecting multiple evaluation indicators. At the same time, we also apply the model to a speech sentiment analysis task. On the CH-SIMS and MOSI datasets, the model shows excellent generalization performance.

## 1. Introduction

As a cross-ethnic and cross-cultural feature shared by all mankind, emotion plays a significant role in the process of human perception and decision-making, especially in increasingly popular human–computer interaction (HCI) systems [1]. Sentiment analysis is a significant mission in natural language processing (NLP), where the main purpose is to make machines understand human emotions [2]. As the main method of people’s daily communication, speech contains abundant emotional information [3]. As an important branch of emotion computing, an SER system can be defined as a set of methods for processing and classifying speech signals to detect implicit emotions [4]. Therefore, how to extract key emotional information from speech signals is a challenging research matter within the scope of speech processing, which has attracted the attention of many relevant researchers.

In order to clearly perceive the emotional changes in speech, extracting the most relevant acoustic features has always been a subject of intense interest in the research of speech emotion recognition [5]. Over the years, sentiment recognition models have gone through significant improvement with the introduction of deep neural networks. In the emotion classification of speech emotion recognition, a common method uses discrete states to represent emotions, for instance, sadness and happiness. Some researchers believe that there is a certain coherence to how people express their emotions in their daily lives. Therefore, three variables of arousal, potency, and valence are used to express people’s emotions in a three-dimensional (3D) continuous space [6].

Commonly used methods of SER mainly include traditional classification or regression algorithms and deep learning methods. Traditional methods have lower algorithm complexity and processing speeds. Commonly used methods include the support vector machine (SVM), the Gaussian mixture model (GMM), the hidden Markov model (HMM), random forest (RF), etc. The SVM aims to find the hyperplane with the largest interval in the sample space to produce more robust classification results. For more complex samples, it can be mapped from the original space to a higher dimensional space. The GMM classifies and converts data into a probabilistic model through unsupervised learning, which provides guidance for many subsequent methods. The HMM can estimate and predict unknown variables based on some observed data. Not only that, but the HMM can also efficiently improve the matching degree between the evaluation model and the observation sequence. RF has a simple structure and a small amount of calculation. It can be used for both classification and regression problems. Even if the dataset is not complete, RF can maintain high classification accuracy. With the advancement of computing equipment and sensor levels, deep learning methods are being chosen by more researchers. Compared with some classic speech emotion recognition models, deep neural networks have greater advantages in terms of training speed and recognition performance.

Deep learning methods can be used to extract speech’s emotional features more accurately. In recent years, many speech emotion recognition models have incorporated deep learning. ED-TTS [7] utilizes speech emotion diarization and speech emotion recognition to model different levels of emotions. This method can obtain fine-grained frame-level emotional information. Zhou et al. [8] proposed an end-to-end speech emotion recognition system based on multi-level acoustic information. The model uses MFCC, spectrogram, and embedded high-level acoustic information to construct a multimodal feature input. Then, it uses a co-attention mechanism for feature fusion. A neural network mainly models the observation probability of speech. Compared with the traditional support vector machine model, Stuhlsatz et al. [9] adopted a DNN network architecture to study discriminative representations from traditional statistical feature Log-Mel spectrograms and achieved higher accuracy. At the same time, due to the excellent performance of end-to-end networks in the image domain, more and more researchers are trying to apply neural networks to the field of speech emotion recognition to extract speech recognition features and their representations [10]. Kim et al. [11] employed CNNs for a new classification study, which applied CNNs to sentence-level classification tasks and achieved good results. In addition, to ameliorate the performance of the current models in emotion analysis and problem classification, they suggested that CNN architecture should be simply modified to allow for the simultaneous use of task-specific and static vectors. Badshah et al. [12] used a method that included CNN architecture with rectangular filters and demonstrated its effectiveness in smart medical centers. In speech emotion recognition, a recurrent neural network (RNN) is also proven to be an effective method to solve emotion analysis problems. Haşim Sak et al. [13] used LSTM, which is a recurrent neural network architecture and achieved good results. Fei et al. [14] introduced an Advanced long short-term memory (A-LSTM) method, which employs a pooled RNN to learn sequences and has a better performance than simple LSTM. Since the attention mechanism has specific characteristics that make the model focus more on the output, it has been extensively verified in many fields [15]. Chung-Cheng Chiu et al. [16] introduced a multi-head attention (MHA) method to improve the existing automatic speech recognition (ASR) framework. In reference [17], to improve the recognition performance of the framework, they used multi-head attention to pay attention to the information from different locations in the subspace.

A spectrogram is a time-frequency decomposition of a voice signal, which shows its frequency content changing with time. In this paper, we focus on the current work of many researchers on deep learning and spectrograms in speech processing. A novel speech emotion recognition architecture is proposed, which is based on multi-head attention and a bidirectional gated recurrent unit (Bi-GRU) network. For speech data, it is challenging to evaluate and compare different methods because there is not enough of a comprehensive public corpus of labeled emotional speeches [18]. We selected the IEMOCAP [19] and Emo-DB [20] corpora to evaluate our proposed model. These corpora contain a lot of data about speech, which are marked as single sentences and used in some advanced research [21].

In the whole research process of speech emotion recognition, emotion recognition from quasi-linguistic components of speech has always attracted a lot of research interest [10,11]. Common sentiment classification research mainly focuses on the classification and regression of feature extraction methods, including short-term frame-level information and discourse-level information [16,17]. In recent research on feature information recognition and classification, compared with the traditional manual low-frame (frame-level) features, in order to improve the classification accuracy of the entire network, the researchers performed statistical learning on the speech features at all levels of the deep network. At present, most speech emotion recognition research mainly uses an architecture composed of neural networks, which includes CNN, RNN, LSTM, or their combination, etc. [22].

As a typical neural network, CNNs are often designed to process data with a grid-like topology, for instance, voice recognition, image processing, etc. By applying correlation filters, CNN can obtain more temporal and spatial feature information from the input signal. Then, it uses its own structure to simplify the input’s original signal without losing the feature form, which effectively improves the classification performance while reducing the complexity of the algorithm [23]. Trigeorgis et al. [24] used CNN to preprocess the raw samples, which can obtain more audio feature information that is beneficial for classification. The experimental results show the advantages of their proposed method in the field of speech recognition. In order to find more feature information that is beneficial to the classification results, Mao et al. [25] proposed a strategy of using CNN to learn support vector machines and achieved good results. In addition to the CNN network, RNN has shown great capacity in many sequence modeling tasks [26]. Kerkeni et al. [27] put forward a method of speech emotion recognition. In their work, MFCC is extracted by using modulation spectrum (MS) features, and then seven emotions are classified from Emo-DB and Spanish databases by using the RNN learning algorithm. Zhao et al. [28] proposed a new CNN–LSTM structure. They used the Log-Mel spectrogram as the input of the entire model to extract the feature information with time series in the speech. In addition, compared with DNNs, Tara N. Sainath et al. [29] found that CNN and LSTM achieve better classification results in a wide variety of speech recognition tasks. Chen et al. [30] proposed a multi-scale fusion framework, STSER, based on speech and text information. They took the log-spectrogram as the input of the model and used the neural network (CNN, Bi-LSTM) and attention mechanism to train and classify the source data.

Li Yang [31] put forward a new emotion analysis frame, SLCABG, which is used to analyze the emotions of a large number of customers on e-commerce platforms when they review products. The model is based on an emotion dictionary, which combines a convolutional neural network and attention-based Bi-GRU to achieve better recognition performance. Over the past few years, the attention mechanism of deep learning has been successful in the context of speech emotion recognition [32,33,34], which prefers to focus on emotionally charged words in people’s communication. Feng Xuanzhen and Liu Xiaohong [35] designed a Bi-GRU and attention-based structure to focus on fine-grained emotional features that are easily ignored by humans, which aims to capture the scattered emotional feature information in comment texts. Po-Yao Huang et al. [36] proposed an attention mechanism training model with a variable number of heads with reference to visual object detection technology. This model is mainly used to obtain feature information from the speaker’s voice, face, and other aspects. With the excellent performance of the end-to-end network in speech recognition, Tomoki Hayashi [37] introduced it into the multi-head attention model to expect better recognition results.

In general, the problems faced by the current SER include the following: (1) Inconsistent feature scales, making it difficult to balance the importance of local features and long sequence emotional features; (2) redundant information limits the accuracy and stability of SER; and (3) using an overly complex model leads to overfitting in performance on a specific task or dataset.

Based on the above factors, we propose a speech emotion model. The main contributions of this paper can be summarized as follows:We propose a new SER model, which combines double Bi-GRU and multi-head attention to extract a spectrogram from speech data to learn speech representation.During training, CNN layers are first applied to study the local associations from spectrograms, and dual GRU layers are used to learn the long-term correlations and contextual feature information. Second, multi-head attention is used to focus on the features related to emotions. Finally, the softmax layer is used to output various emotions to improve the overall performance. After experimental verification, our proposed model has better classification performance, and its unweighted accuracies on person-neutral IEMOCAP and Emo-DB sentiment datasets reached 75.04% and 88.93%, respectively. Different from the methods proposed in references [38,39], our proposed model achieves better classification performance in speech emotion recognition. Moreover, the gated recurrent unit (GRU) is more efficient than long short-term memory in training [40].We also conducted training on different tasks and datasets to verify the generalization performance and stability of the model. Finally, we analyzed and summarized the results of the experiment and proposed possible future research points.

## 2. Materials and Methods

In this section, a new speech emotion recognition model is proposed, which combines the Bi-GRU network and attention mechanism for the extraction of speech signal emotion features. Figure 1 shows the network structure we constructed, mainly including the spectrogram feature input, gated recurrent unit, multi-head attention layer, and softmax layer, which are used for the extraction, training, and classification of speech emotion information. The Bi-GRU layer is used to learn long-term correlation and contextual feature information. Multi-head attention is used to focus on emotion-related features. The softmax layer is used to output various emotions to improve the overall performance.

### 2.1. Spectrogram Extraction

As described in [38], the spectrogram has better performance in speech emotion recognition. In the preparation stage of our experiment, since the spectrogram can retain rich frequency domain information, we adopted the spectrogram as the input information of the model in the proposed speech emotion model. We obtained the spectrogram by taking the short-time Fourier transform of the original speech signal. To preserve the integrity of emotional information in the original speech signal, we sampled the speech signal in the experimental dataset at 16 KHz and organized it into a single sentence with a duration from less than a second to about 20 s. In our experiments, we use the IEMOCAP and Emo-DB corpora and select a similar-sized sentiment subset. This approach can help us comprehensively analyze the classification performance of the model in the training process for each type of emotion. In order to more intuitively explain the proportion of each emotion in the corpus in the experiment, we filled in the number of sentences selected for each emotion information in Table 1. Every sentence is marked with at least one emotion. With respect to the overlapping hamming window sequences, we set the frame step size to 10 ms and the frame length to 40 ms. For each frame in training, the specific calculation length of the DFT in the experiment is 1600. In other words, we used 10 Hz grid resolution. Since the frequency of the human voice signal is generally 300–3400 Hz, and after repeated verification and analysis of the experiments, the frequency range of 0–4 KHz was finally selected, and others were ignored. After summarizing the short-time spectrum, we obtained a matrix with the size of (N × M); in specific experiments, the variable N generally corresponds to the number of speech sentences and is used to represent the selected temporal grid resolution. The variable M was used to represent the frequency grid resolution in the experiment. After obtaining the DFT data, we converted it into a logarithmic power spectrum. Then, the logarithmic power spectrum was normalized by z using the mean and standard deviation of the training dataset.

Sentence lengths of speech samples are usually different. In order to upgrade the computational efficiency, we sort by length. If there are spectrograms with a similar timing length in the input sequence, we will organize them into the same batch and pad with 0 to the maximum length of spectrograms in the current batch. In the training stage of our proposed network, we performed unified parallel calculations on a batch of samples.

### 2.2. The Bi-GRU Layer

The GRU network, as another form of LSTM, was proposed to solve problems such as long-term memory and the gradient in backpropagation. Compared with the LSTM network, it can not only obtain the contextual feature information of the input speech, but it also has lower computational complexity. It is well known that speech signals are closely related to the time dimension. In the process of conversation, people usually think about the content of a conversation in a future time based on the utterance information of the previous time. In our work, the GRU network layer also takes advantage of its advantages in the time dimension. In other words, the experiment will first analyze the emotional tone of the whole utterance and then deeply analyze the local emotional information features in the utterance. Regarding the study of the GRU network, Cho et al. [41] set up the same experimental conditions in their study to test the GRU and LSTM network models. The experimental results show that the algorithm complexity of the GRU network is smaller in training, and the recognition performance of the model is also better [8].

For the traditional GRU network, the emotional features in the speech are usually extracted and analyzed according to the time rule, and the potential emotion of the next moment is determined by analyzing the emotion of the previous moment. However, it has been found in practice that emotional changes in people’s daily conversations are quite complex, and the current speech may also be related to future speech. For example, humans express current words with a certain emotion, and the emotion is constantly changing as the conversation progresses. Considering the above situation and the research and verification in the experiment, if the Bi-GRU layer is added to the whole network, it can better compensate for part of the emotional feature loss caused by the single-item GRU network. In a typical GRU network architecture, the architecture mainly includes a reset gate and an update gate. Intuitively, the reset gate determines how the new input information is combined with the previous memory, and the update gate defines the amount of previous memory saved to the current time step. Quantitative calculations are shown in Equations (1) and (2). When $x_{t}$ is input at time *t*, the bidirectional GRU network can obtain $h_{t}$ and $\tilde{h_{t}}$, which are the hidden states of forward and reverse information, respectively.
(1)$$\vec{h_{t}}=\vec{GRU}\left(x_{t},{\vec{h}}_{t-1}\right)$$
(2)$$\overset{\leftarrow }{h_{t}}=\overset{↼}{GRU}\left(x_{t},{\overset{↼}{h}}_{t-1}\right)$$


Both forward and backward propagation of the Bi-GRU include the calculation of the reset gate $r_{t}$, update gate $z_{t}$, and the hidden state $\tilde{h}$. Their calculation processes are as follows:(3)$$r_{t}=\sigma \left(W_{r}\cdot \left[x_{t},h_{t-1}\right]+b_{r}\right)$$
(4)$$z_{t}=\sigma \left(W_{r}\cdot \left[x_{t},h_{t-1}\right]+b_{z}\right)$$
(5)$$\tilde{h}=tanh\left(W\cdot \left[x_{t},r_{t}⨀h_{t-1}\right]+b\right)$$
where σ represents sigmoid, *W* represents weight, and *b* represents bias.

The hidden state outputs at time *t* are $\vec{h_{t}}$ and $\overset{\leftarrow }{h_{t}}$, where $h_{t}=\left[\vec{h_{t}};\;\overset{\leftarrow }{h_{t}}\right]$. Then, the global context information is obtained by combining $\vec{h_{t}}$ and $\overset{\leftarrow }{h_{t}}$ to capture the speech context feature information vector.

### 2.3. The Multi-Head Attention

Over the past few years, since the attention mechanism of deep learning networks can ensure that the classifier pays attention to the specific position of a given sample according to the attention weight of each part of the input, it is often applied in the field of speech emotion recognition. The attention function mainly includes these vectors, namely query, key, value, and output. Generally speaking, a typical attention mechanism automatically learns and calculates the contribution of the input data to the output data by computing the mapping of keys and values. Generally speaking, a typical attention mechanism automatically learns and calculates the contribution of the input data to output data by computing the mapping of keys and values. During training, the model predicts the current time step based on the input data and the historical output of neurons and finally obtains the weights of each dimension of the input data. Each value in this calculation process needs the compatibility function of the query and the matching key to calculate. The method adopted by the traditional attention mechanism is to concentrate the acquired context vector on the specific representation subspace of the input sequence. However, the context vector obtained in this way only reflects one aspect of the semantics in the input. Generally speaking, a sentence may involve multiple semantic spaces, especially for a long input sequence.

In order to achieve better classification performance during model training, a multi-head attention mechanism is introduced into our Bi-GRU model. The multi-head attention model can represent different subspaces according to the number of heads and obtain common attention information in different positions.
(6)$$MultiHead\left(Q,K,V\right)=Concat\left(head_{1},\ldots ,head_{h}\right)W^{O}$$
(7)$$head_{i}=Attention\left(QW_{i}^{Q},KW_{i}^{K},VW_{i}^{V}\right)$$
where the projections are parameter matrices $W_{i}^{Q}\in R^{d\times d_{q}}$, $W_{i}^{K}\in R^{d\times d_{k}}$, $W_{i}^{V}\in R^{d\times d_{v}}$, and $W_{i}^{O}\in R^{hd_{v}\times d_{1}}$.

In this work, after obtaining the output vector of the GRU network, we find it beneficial to linearly project the $d_{q}$, $d_{k}$, and $d_{v}$ dimensions for the *h* times. This linear projection is obtained by different and learned ways of query, key, and value, respectively. Then, we execute the attention function in parallel through the projection of each query, key, and value and generate the output value of the $d_{v}$ dimension. Finally, these vectors are connected and projected again to obtain the final values.

The overall process of the model is shown in Algorithm 1.

**Algorithm 1.** The pseudocode of the model.**Input:** Time-frequency characteristics X1,X2,…,XT**Output:** Emotion categories and probabilities *Y, P*// Bi-GRU algorithm (forward and backward)ht→←GRU→xt,ht−1ht↼→GRU↼xt,h˜t−1ht=[ht→,ht↼]// Multi-head Attention algorithmQ=ht⋅Wq, K=ht⋅Wk, V=ht⋅WvAttentionScorei=softmaxQWQiTKTdkheadi=AttentionScorei⋅VMultiHeadQ,K,V=Concathead1,…,headhWO// Linear layer and output layer Y′=full connectMultiHeadQ,K,VY=softmax(Y′)

## 3. Results

As introduced in previous sections, in terms of the dataset selection, we chose the IEMOCAP and Emo-DB datasets to evaluate our proposed system. At the same time, in order to verify the scalability of the model for different tasks, we also applied the model to the speech emotion analysis task. The datasets selected for the speech sentiment analysis task are CH-SIMS [42] and MOSI [43].

### 3.1. Experiment Setup

The IEMOCAP [19] datasets are mainly used to study the multimodal expression of binary interactions. This corpus is mainly composed of facial motion capture and the associated audio/video recordings of 10 subjects within 12 h. Each session consists of a different two-person group. Actors perform scripted plays and engage in spontaneous improvised dialogs elicited through affective scenario prompts.

We chose the EMO-DB [20] dataset as another speech signal sample input for this experiment. The main reasons include the diversity of voice characters, emotion types, and its broadness in the field of speech emotion recognition. In this way, we can better complete the comparison with other previous research work. The Emo-DB corpus contains seven emotional categories as follows: anger, sadness, fear/anxiety, neutrality, happiness, disgust, and boredom, which together constitute 535 audio utterances in German.

The leave-one-speaker-out (LOSO) method was used to conduct the experiments. In each model training test, all the speeches of one person were used as the test set, while the rest of the others’ speeches were used as the training set. This method can more accurately verify whether the algorithm can learn the characteristics of emotion, and this feature is independent of the speaker. During training, in order to increase the accuracy of sentiment classification in the experimental results and reduce the probability of bias caused by accidental events, we used the cross-validation method in every experiment. In the experiment, we selected N people’s speech samples as the test sets and then calculated the statistical average of the results as a complete experimental result.

In this paper, we built and trained all the models based on the NVIDIA GeForce RTX 3060Ti GPU platform and the python3.9 + pytorch1.12.1 + cuda11.6 deep learning framework. In our experiments, the Adam optimizer was used to optimize the model’s parameters. The loss function of the model uses L1 loss, and the activation function uses ReLU. With differences between the IEMOCAP and Emo-DB datasets in mind, we set two sets of hyperparameters, respectively, as shown in Table 2. For the settings of hyperparameters, we followed the common values used by researchers. For the IEMOCAP dataset, we set a lower learning rate and a higher number of attention heads and epochs. This is because the data in the IEMOCAP dataset are larger. At the same time, in order to reduce the phenomenon of overfitting, we also increased the dropout record to 0.3.

### 3.2. Experimental Results

We evaluated five groups of emotion recognition experiments by reporting the unweighted accuracy (UA), which is the accuracy of all samples in the test set. As a classifier used to model the sentiment features, two fully connected layers and a softmax layer were connected after the multi-head attention layer. For the rest, we adopted the model of the Bi-GRU combined with multi-head attention. In the experiment, the experiment set number of heads was 4, 8, 16 and 32, respectively. The purpose of this setting is to explore the influence of the head number on emotion recognition in multi-head attention. In addition, we will only use the Bi-GRU network model in Table 3 as the experimental baseline.

In order to analyze the whole experimental process more comprehensively, the confusion matrix of our network (head = 16) for the IEMOCAP corpus is given in Figure 2, and the confusion matrix of our network (head = 8) for the Emo-DB corpus is given in Figure 3. For the speaker-independent IEMOCAP and Emo-DB datasets, the new model has a good recognition rate for neutral and sad emotions. Because there are more neutral emotion samples in the corpus, the training effect is better. Compared with other emotions, sadness has more obvious emotional characteristics on the spectrogram. Compared with other emotional samples, happy samples and angry samples have more similar emotional features in the IEMOCAP dataset, which achieves the lowest recognition effect. In the Emo-DB dataset, fear, upset, and bored emotions also obtain higher recognition rates.

In Table 4, we can see that the unweighted accuracy (UA) obtained during model training is not exactly the same for different corpora. When the head numbers are 16 and 8, the best performance is 75.04% and 88.93% in the speaker-independent audio classification tasks, respectively.

In order to show the overall performance of the model more comprehensively, we calculated the overall accuracy, precision, and recall in the two datasets, IEMOCAP and EMO-DB, as shown in Table 4.

The recognition accuracy of different emotions is shown in Table 5.

At the same time, we compare the proposed model with commonly used deep learning methods and discuss the experimental results in Section 4. The experimental results are shown in Table 6.

In order to explore the generalization performance and scalability of the proposed model, we also conducted experiments on speech sentiment analysis on the CH-SIMS and MOSI datasets. The experimental results of the model in the speech emotion analysis task are shown in Table 7. The evaluation indicators of the experiment include binary accuracy (Acc-2), five-class accuracy (Acc-5), the F1-score, mean absolute error (MAE), and Pearson’s correlation coefficient (Corr). For all metrics except MAE, higher values indicate a better performance. We analyze the extended experimental results in Section 4.

## 4. Discussion

According to Table 4, it can be concluded that the model can show relatively good performance with both datasets. Judging from the results, the accuracy of negative emotion recognition by IEMOCAP and Emo-DB is higher than the accuracy of positive emotions. We analyzed that this may be because negative emotions have more obvious characteristics in speech, while positive emotions are more easily recognized as neutral emotions.

As can be seen in Table 6, our proposed method has a higher accuracy compared with commonly used deep learning models. This is due to the joint effect of Bi-GRU and multi-head attention in the model, which allows the model to pay attention to the overall and local characteristics of the speech sequence. Our model combines the advantages of attention and recurrent neural networks. Compared with independent models such as Bi-GRU, self-attention, and multi-head attention, our model complements and fuses information to make the results of speech emotion recognition more stable.

According to Table 7, the model can also achieve relatively good results in the speech emotion analysis task. This is because the Bi-GRU layer and multi-head attention layer in the model fully extract the information in the speech signal and adopt a reasonable dropout probability to increase randomness.

The analysis of variance (ANOVA) statistical test was used to evaluate whether there are differences between classes. The model based on Bi-GRU and multi-head attention showed statistically significant improvements over the Bi-GRU network individually for both the Emo-DB $\left(F\left(1,698\right)=10.0243,\;p<0.05\right)$ and the IEMOCAP $\left(F\left(1,198\right)=8.4174,\;p<0.05\right)$.

In Figure 4, with the ever-changing headcount, the accuracy of the speech emotion recognition model tends to rise first and then decline. When the number of heads increases, the complexity of the model itself increases. The emotion that the model pays attention to becomes more detailed, which makes it deviate when learning weights.

## 5. Conclusions

This paper proposes a new speech emotion recognition model, which combines the GRU network and multi-head attention mechanism. On IEMOCAP and Emo-DB datasets, the proposed model achieves 75.04% and 88.93% unweighted accuracy, respectively. Compared with the results in [38], our results are a 10.82% improvement on the IEMOCAP dataset. At the same time, compared with the results obtained by Mustaqeem et al. [39], the experimental results were improved by 2.79% and 3.36% on the IEMOCAP and Emo-DB datasets, respectively. In addition, the model has been improved by 4.40% and 5.71%, respectively, based on the baseline (Bi-GRU) [44]. Compared with most advanced speech emotion recognition models, our model has a better performance on the IEMOCAP and Emo-DB datasets. We further compared the proposed model with common deep learning methods, and the emotion recognition model based on Bi-GRU and multi-head attention methods achieved the best recognition results. In addition, we also applied the model to the speech emotion analysis task and achieved more stable prediction results on different datasets. In these experiments, we used rich evaluation indicators to conduct a more comprehensive analysis of the model and prevent the model from overfitting for specific parameters. Finally, we analyzed the results of the experiment.

SER is of great significance in emotion recognition. Due to the variability of emotions, a piece of speech often has multiple emotions, which is challenging for the accurate extraction of speech information features. For multiple languages, cross-cultural emotion recognition is the future development trend. People in different countries and regions have certain cultural differences, but for humans, even if they cannot understand what foreigners are saying, they can roughly understand their tone and attitude. In the future, the cost of training the model may be interesting, especially on long input sequences. The idea of future work is similar to that of Nikita Kitaev [45]. In order to reduce the performance loss of the model in the multi-head attention layer algorithm, we will add locality-sensitive hashing (LSH) attention to our subsequent research. 

It can be seen from the experimental results that the recognition accuracy of our model for a specific emotion has not been greatly improved. We analyzed that this may be due to the fact that the speech features of the emotion of calm are not obvious, and our model does not use complex methods in feature fusion. In the future, we will consider conducting research on feature fusion. At the same time, speech is only one aspect of emotion. Combining text information with speech recognition will significantly improve the performance of emotion recognition. This research direction is also more in line with the development of artificial intelligence.

At the same time, multimodal sentiment analysis [46] is also very popular. It has richer emotional information and is more comprehensive in predicting and identifying human emotions. Therefore, our model can be improved and applied to more scenarios with the help of advanced methods such as multi-task learning methods, transfer learning, or joint learning.

## Figures and Tables

**Figure 1 sensors-24-01429-f001:**
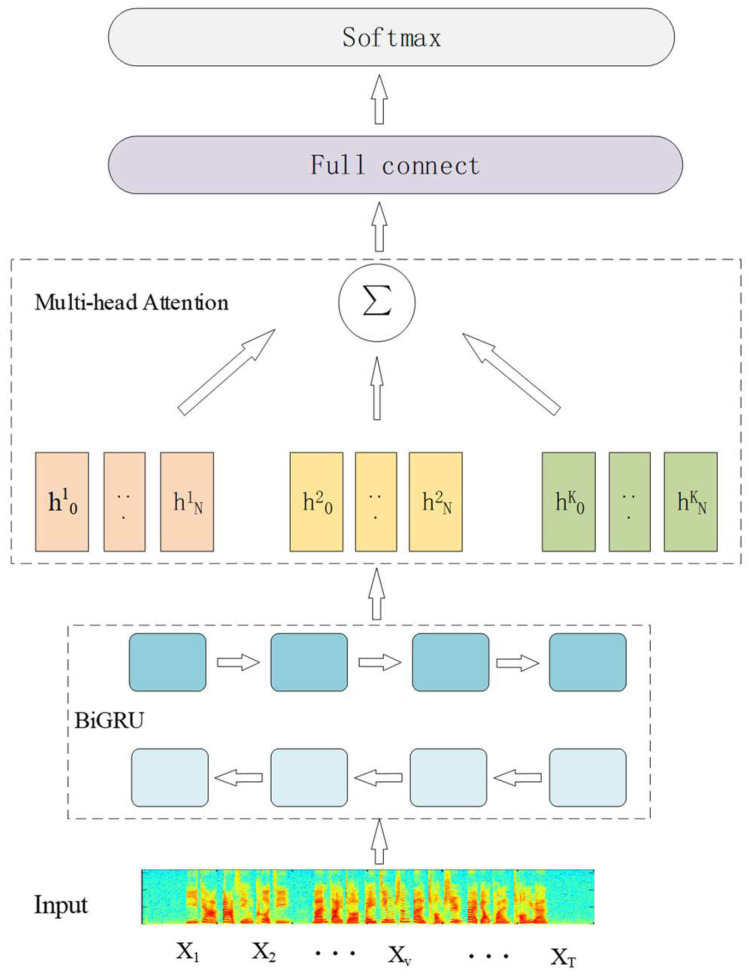
A network structure based on Bi-GRU and multi-head attention.

**Figure 2 sensors-24-01429-f002:**
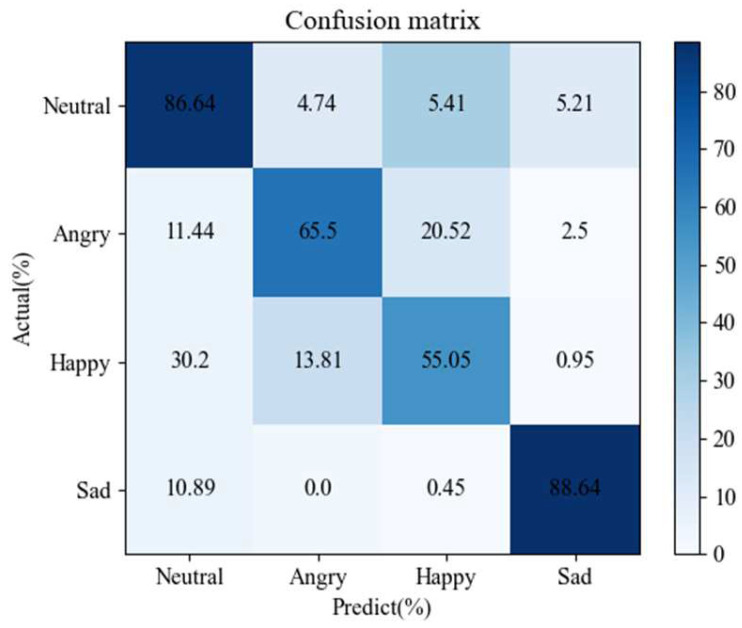
The confusion matrix of our network (head_num = 16) for the IEMOCAP corpus.

**Figure 3 sensors-24-01429-f003:**
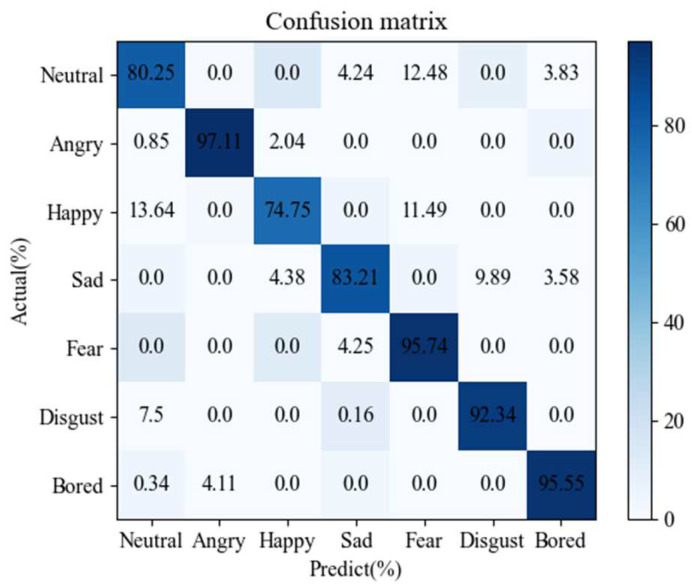
The confusion matrix of our network (head_num = 8) for the Emo-DB corpus. The confusion matrix of our network (head_num = 16) for the IEMOCAP corpus.

**Figure 4 sensors-24-01429-f004:**
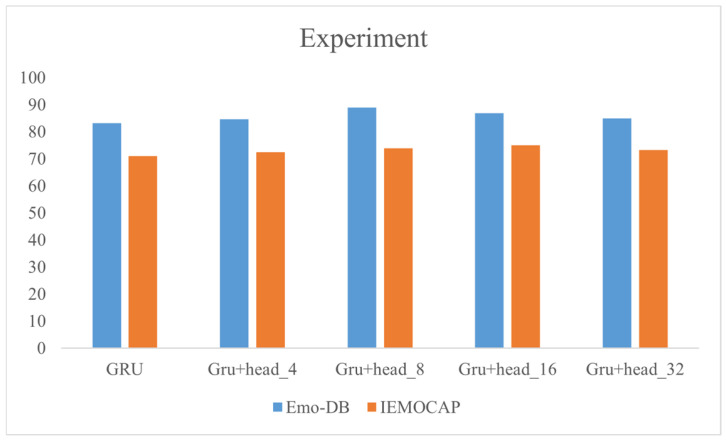
Line chart of unweighted precision (UA) on IEMOCAP and Emo-DB datasets.

**Table 1 sensors-24-01429-t001:** Details of IEMOCAP and Emo-DB corpora.

Emotion	Emo-DB	IEMOCAP
neutral	50	50
angry	50	50
happy	50	50
sad	50	50
fear	50	-
disgust	50	-
bored	50	-

**Table 2 sensors-24-01429-t002:** Hyper-parameter settings of the model.

Hyper-Parameter	IEMOCAP	Emo-DB
Learning rate	1 × 10^−5^	1 × 10^−4^
Batch size	32	32
Att dropout	0.2	0.2
Head num	16	8
Dropout (Output)	0.3	0.2
Epochs	100	80

**Table 3 sensors-24-01429-t003:** Comparison of unweighted accuracy (UA) on IEMOCAP and Emo-DB datasets.

Model	IEMOCAP	Emo-DB
Bi-GRU	71.01%	83.22%
Head num = 32	73.32%	85.03%
Head num = 16	75.04% *	86.81%
Head num = 8	73.89%	88.93% *
Head num = 4	74.47%	84.66%

In the Emo-DB and IEMOCAP corpus, the * indicates that there are significant differences between the estimated values of the Bi-GRU model and the proposed model.

**Table 4 sensors-24-01429-t004:** Details of model results on IEMOCAP and Emo-DB datasets.

Results	IEMOCAP	Emo-DB
Precision	83.33%	74.57%
Accuracy	75.04%	88.93%
Recall	70.20%	80.20%

**Table 5 sensors-24-01429-t005:** Recognition accuracy of different emotion categories on IEMOCAP and Emo-DB datasets.

Emotion Category	IEMOCAP	Emo-DB
Neutral	80.25%	86.64%
Angry	97.11%	65.50%
Happy	74.75%	55.05%
Sad	83.21%	88.64%
Fear	95.74%	-
Disgust	92.34%	-
Bored	95.55%	-

**Table 6 sensors-24-01429-t006:** Accuracy of different models on IEMOCAP and Emo-DB datasets.

Method	IEMOCAP	Emo-DB
CNN	69.80%	70.58%
LSTM	70.10%	77.73%
RNN	70.22%	79.91%
CNN + LSTM	72.17%	83.79%
Bi-GRU	71.01%	83.22%
Self-Attention	71.87%	82.91%
Bi-GRU + Self-Att	73.21%	84.58%
Multi-head Attention	72.13%	83.37%
Our Method	75.04%	88.93%

**Table 7 sensors-24-01429-t007:** Experimental results of the model in speech sentiment analysis task.

Evaluation Index	CH-SIMS	MOSI
Acc-2	72.43%	70.29%
F1-score	71.97	70.31
MAE	0.648	0.974
Corr	0.524	0.613
Acc-5	30.19%	32.12%

## Data Availability

Some of the datasets mentioned in the paper can be downloaded at https://sail.usc.edu/iemocap/ accessed on 1 October 2023, http://emodb.bilderbar.info/docu/ accessed on 16 June 2022.

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
