# Peer review of "A New Network Structure for Speech Emotion Recognition Research"

_sensors, 2024, doi:10.3390/s24051429_

Round 1
Reviewer 1 Report
Comments and Suggestions for Authors
Title: “A new network structure for speech emotion recognition research”
The paper under consideration explores the application of a network model based on Gated Recurrent Unit (GRU) and multi-head attention for speech emotion recognition (SER). The study emphasizes the significance of acoustic feature extraction, particularly from spectrograms, and evaluates the proposed model on the Emo-DB and IEMOCAP corpora.
The authors presented that the incorporation of GRU and multi-head attention into their network model leads to improved results compared to traditional models, as evidenced by superior unweighted accuracy (UA) on the evaluated datasets.
While the paper addresses an area of considerable interest and relevance, it is essential to critically examine several aspects of the study.
Strengths:
- Focus on spectrograms: Recognizing the importance of spectrograms for capturing linguistic information in speech emotion recognition (SER) is a positive step. Spectrograms visually represent speech frequencies and their intensities over time, making them rich in features relevant for emotion detection.
- GRU and multi-head attention: Utilizing GRU cells, known for their effectiveness in handling sequential data like speech, and multi-head attention mechanism for focusing on relevant parts of the input are promising choices for SER tasks.
- Comparative evaluation: Comparing the proposed model with traditional models on established datasets like IEMOCAP and Emo-DB provides a good basis for evaluating its performance.
- Reported improvement: Achieving higher unweighted accuracy (UA) than traditional models suggest potential effectiveness of the proposed network.
Weaknesses:
- Missing details: The paper lacks crucial details about the proposed network architecture. The specific configuration of the GRU and multi-head attention layers, activation functions, loss function, and optimizer used are all missing. This information is essential for understanding the model's inner workings and comparing it fairly with other approaches.
- The study evaluates the proposed model on the Emo-DB and IEMOCAP corpora. While these datasets are widely used in emotion recognition research, the generalization of the findings to diverse datasets remains unclear. A more extensive evaluation across multiple datasets would strengthen the external validity of the proposed model.
- Limited evaluation metrics: Relying solely on UA as the evaluation metric does not provide a comprehensive picture of the model's performance. Including metrics like precision, recall, confusion matrix, and class-specific accuracies would give a better understanding of how the model behaves for different emotions.
- Qualitative analysis missing: Analyzing the model's predictions and interpreting what features it might be focusing on for emotion recognition would enhance the understanding of its strengths and limitations.
Suggestions for improvement:
- Provide detailed specifications of the network architecture, including layer configurations, activation functions, loss function, and optimizer.
- Expand the evaluation metrics to include precision, recall, confusion matrix, and class-specific accuracies.
- Perform qualitative analysis of the model's predictions to understand which features it focuses on for different emotions.
- The introduction could be strengthened by providing a more nuanced overview of existing challenges and limitations in SER, specifically related to feature extraction and network architectures.
- The conclusion could be expanded to discuss the potential applications of the proposed model and future research directions.
By addressing these suggestions, the authors can enhance their paper's scientific rigor and potential impact on the field of speech emotion recognition.
Comments on the Quality of English LanguageMinor Changes are required i.e., Grammar and spelling Check
Reviewer 2 Report
Comments and Suggestions for Authors
The paper presents a novel network model for speech emotion recognition (SER) that integrates a Gated Recurrent Unit (GRU) and multi-head attention mechanism. It achieves higher unweighted accuracy (UA) on Emo-DB and IEMOCAP datasets compared to traditional models.
But I have some concerns for this paper:
1) The increased model complexity with more attention heads may lead to overfitting or difficulty in learning weights.
2) Relies on specific datasets (Emo-DB and IEMOCAP), which may not generalize to other types of speech data.
3) Potential high computational cost for training, especially on long input sequences, which may limit practical applications.
There also lacks some related works such as:
1) T. M. Wani, T. S. Gunawan, S. A. A. Qadri, M. Kartiwi and E. Ambikairajah, "A Comprehensive Review of Speech Emotion Recognition Systems," in IEEE Access, vol. 9, pp. 47795-47814, 2021, doi: 10.1109/ACCESS.2021.3068045.
2) Kaur, Kamaldeep, and Parminder Singh. "Trends in speech emotion recognition: a comprehensive survey." Multimedia Tools and Applications (2023): 1-45.
3) Wu, Zhen, Yizhe Lu, and Xinyu Dai. "An Empirical Study and Improvement for Speech Emotion Recognition." ICASSP 2023-2023 IEEE International Conference on Acoustics, Speech and Signal Processing (ICASSP). IEEE, 2023.
Reviewer 3 Report
Comments and Suggestions for Authors
The article titled " A new network structure for speech emotion recognition research" proposes a speech emotion recognition model that combines Bidirectional Gated Recurrent Unit (Bi GRU) and multi head attention mechanism. The research focuses on extracting emotional features from spectrograms and evaluating them using the IEMOCAP and Emo DB datasets. The model outperforms traditional models in unweighted accuracy (UA). After careful review, I believe that this article lacks innovation and the experimental design is relatively weak. There are specific issues as follows:
1、 abstract
The abstract writing is too objective and vague. It is recommended to highlight the main problems to be solved and how to solve them, and highlight the innovation of the solution methods.
2、 introduce
It is recommended to systematically classify and summarize existing research, and compare and discuss the proposed methods with recent work, in order to highlight the novelty and contribution of our method.
3、 Insufficient detailed description of the model
Although this article proposes a new model, the description in parameter selection and network structure design is not specific enough, and the theoretical support is relatively weak. Suggest supplementing the content in Chapter 2.
4、 Experiments
The experimental design is too simplistic. It is recommended to increase the number of experiments, introduce more diverse datasets for testing, compare and analyze with other advanced models, or conduct in-depth analysis of instances of model misclassification.
5、 conclusion
l In the conclusion, it would be valuable to emphasize the need for further empirical research and practical experiments to validate the findings and overcome the limitations identified.
l I suggest the author discuss future research methods and limitations of this paper in the conclusion.
6、 reference
The references are relatively outdated, it is recommended to add some relevant achievements from recent years. And it is necessary to ensure that each cited reference is closely related to the topic of the paper, especially the relevant literature in the experimental design and methodology sections.
7、 Formatting:
l Is the input in Figure 1 complete? It is recommended to check.
l The title of Table 1 has not been changed to your own.
l Abbreviations of nouns only need to be indicated at the first occurrence.
Round 2
Reviewer 2 Report
Comments and Suggestions for Authors
This paper proposes a new network structure for speech emotion recognition research, based on GRU and multi-head attention. The paper introduces a novel speech emotion recognition architecture that can extract key emotional information from spectrograms, and achieve better performance than traditional methods on two public datasets (IEMOCAP and Emo-DB). The paper combines Bi-GRU and multi-head attention to capture both the global and local context of speech signals, and to focus on the relevant features for emotion recognition. The paper also applies the proposed model to the speech sentiment analysis task, and shows its generalization and scalability on two other datasets (CH-SIMS and MOSI).
However, the paper does not compare the proposed model with other state-of-the-art methods that use deep learning and attention for speech emotion recognition, such as [36] and [37]. I also recommended the authors add the review of the related works about speech emotion modeling, such as the related work of ED-TTS, I think the feature during the speech synthesis could help with the speech emotion recognition.
Reviewer 3 Report
Comments and Suggestions for Authors
Comments and suggestions for authors
After careful review and a round of revisions, I believe that this article still has the following issues:
1、Section 1 lines 44‐50 seems to be a summary of existing methods. It is suggested to summarize the existing research of others, which means that the reference in the introduction section is not simply about what research others have done, but rather about what inspiration the authors can draw from their research, and the necessity and feasibility of this study are explained by previous studies.
2、The author still has not summarized the novelty and contribution of the article, and it is unclear from the introduction what improvements or extensions this method has compared to other studies. What factors do "based on the above factors" refer to in line 133 of the article? The author should provide additional explanations on this in the article.
3、The content added by the author in Section 3.1 lines 279-286 did not describe the details of the model. In lines 146-147, the author wrote that a new model, the Bi GRU model, was proposed, but no specific working details of the model were found in Chapter 2 and Chapter 3. It is recommended to add pseudocode or more formulas to indicate how the model was trained and tested.
4、Suggest providing a more detailed explanation of Figure 1.
5、I suggest providing a brief explanation of the parameter settings in Table 2 and explaining why they are set in this way.
6、The setting in Table 4 seems to be not very accurate, for example, the value corresponding to Angry and IEMOCAP is 97.11%. Therefore, the meaning represented by this 97.11% cannot be directly seen in the table. It is recommended to modify it again.
7、Through the experiment and conclusion section, it can be seen that the recognition accuracy of the proposed method is not very high, and compared with the baseline method, the improvement is not much. Therefore, there are still limitations in the recognition accuracy of the algorithm in this article. Please analyze and summarize this, and propose possible research points for future research.
